# Zero energy states clustering in an elemental nanowire coupled to a superconductor

**Lauriane C. Contamin[1,2], Lucas Jarjat ®[1,2], William Legrand ®[1], Audrey Cottet[1], Takis Kontos[1] & Matthieu R. Delbecq ®[1]** ✉

Nanoelectronic hybrid devices combining superconductors and a one-dimensional nanowire are promising platforms to realize topological super-conductivity and its resulting exotic excitations. The bulk of experimental studies in this context are transport measurements where conductance peaks allow to perform a spectroscopy of the low lying electronic states and potentially to identify signatures of the aforementioned excitations. The complexity of the experimental landscape calls for a benchmark in an elemental situation. The present work tackles such a task using an ultra-clean carbon nanotube circuit. Specifically, we show that the combination of magnetic field, weak disorder and superconductivity can lead to states clustering at low energy, as predicted by the random matrix theory predictions. Such a phenomenology is very general and should apply to most platforms trying to realize topological superconductivity in 1D systems, thus calling for alternative probes to reveal it.

Topological superconductivity (TS) is predicted to exist naturally in some exotic *p*-wave superconductors[1]. It is however actively sought for in low-dimensional nanoscale devices where it can be engineered by combining a standard BCS superconductor with 1D conductors having a large Rashba spin-orbit interaction (SOI) under magnetic field[2–4]. These devices would offer tools to manipulate emerging Majoranas and demonstrate their non-abelian character, with prospects of applications for topologically protected quantum computation[5]. It has turned out that combining all the ingredients together is more challenging than anticipated. The expected signature of TS, namely a robust zero bias conductance peak (ZBCP), can also be explained with nontopological states, as confirmed by a series of recent experimental works[6–8]. These experiments were performed in proximitized short InAs or InSb semiconducting nanowires (NW) that have a large intrinsic Rashba SOI. This last ingredient leads to unconventional odd-frequency superconductivity, precursor of TS, which can naturally lead to nontopological ZBCP[9]. In these works, the multiple transverse subbands of the NW and the strong SOI add ingredients of complexity to the electronic spectra. It is thus essential to investigate proximity induced superconductivity in a long NW with weak SOI in order to benchmark the physics of ZBCP.

Carbon nanotubes (CNT) are the closest possible material to ideal 1D conductors. Having a diameter much smaller than the electronic Fermi wavelength ensures that a single transverse subband participates in transport. This contrasts with semiconducting NWs which typically present several subbands whose intercoupling can lead to apparent ZBCP[10,11]. In addition, CNTs have a weak intrinsic SOI that is not of Rashba type[12]. They are therefore ideal candidates to investigate proximity induced superconductivity in a NW that cannot host TS. It is well established that superconductivity can be induced in CNTs[13]. Most studies so far focused on tuning magnetic flux in SQUID geometries, temperature or the chemical potential in CNT quantum dots in the presence of a competition between superconductivity and Kondo correlations[14–18]. Surprisingly, investigations in magnetic field are scarce[19] and performed in the quantum dot limit. We aim to perform our study in a different limit, that is $\delta \lesssim \Delta^*$ with $\Delta^*$ the proximity induced superconducting gap in the conductor. Such a limiting case can be viewed as the 1D limit for superconducting proximity effect since it corresponds to have the length $L$ of the NW to be greater than the proximity induced superconducting coherence length $\xi$. It would be suitable for the emergence of TS and possibly topological modes if our NW had a suitable spin-orbit coupling[20]. Therefore our device

[1]Laboratoire de Physique de l'École normale supérieure, ENS, Université PSL, CNRS, Sorbonne Université, Université Paris Cité, F-75005 Paris, France. [2]These authors contributed equally: Lauriane C. Contamin, Lucas Jarjat. ✉e-mail: matthieu.delbecq@ens.fr

offers a good benchmark of a system that cannot host TS but which would have all the requirements except spin-orbit interaction for the possible emergence of TS.

In this work, we implement a device meeting the criteria discussed above; An ultra-clean carbon nanotube with weak SOI is contacted to a superconducting lead and is in the 1D regime as defined, with $\delta \lesssim \Delta^*$. We observe fluctuations of the proximity induced mini-gap as a function of the charge ground state. These fluctuations, which stem from weak disorder due to the presence of several gates, follow a universal distribution with a crossover from Gaussian orthogonal ensemble to Gaussian unitary ensemble as magnetic field is applied (equivalently as time reversal symmetry is broken), as predicted by random matrix theory. The presence of such mesoscopic fluctuations of the induced mini-gap were predicted to lead to ubiquitous zero energy edge states as magnetic field is applied. We indeed observe the presence of zero energy states at finite magnetic field for multiple charge ground states by performing magneto-spectroscopy transport measurement. It shows that a general mechanism can lead to clustering of non-topological zero energy states mimicking the transport signatures of Majorana modes.

## Results

Our device shown in Fig. 1a and sketched in Fig. 1b fulfills these requirements. A 30 μm long CNT is stapled over Nb(40 nm)/Pd(10 nm) superconducting leads and Al/AlO$_x$ gates. The stapling is done under a high vacuum of about $10^{-6}$ mbar, providing ultra clean CNT devices[21]. The CNT is electrically cut on both sides between the pairs of outer contacts, giving a 5 μm long NW. One of the two superconducting contacts used in the experiment shows a large residual density of states (DOS) at low energy, as we systematically observe[22] and as also reported in InSb devices[23]. Bias voltage $V_{sd}$ is applied to this "normal" (N) contact and differential conductance $G_{diff}$ is measured through the

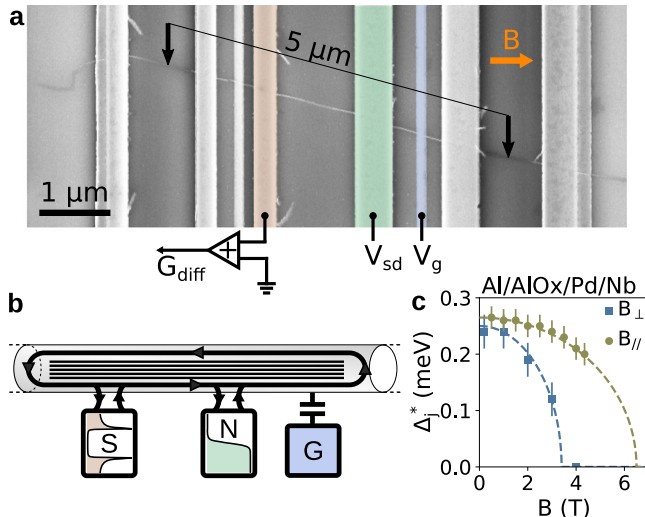

**Fig. 1 | General concepts and experimental implementation. a** False-color scanning electron micrograph of the device. A CNT (white) is stapled on an array of Al/AlO$_x$ gates (thin) and Nb/Pd contacts (large) electrodes. The black arrows indicate where the CNT is electrically cut. Bias voltage $V_{sd}$ is applied to the green contact which behaves as a normal reservoir. The brown contact behaves as a superconducting reservoir and is used as drain to measure the differential conductance through the device. The blue electrode is used as a back gate. External magnetic field $B$ is applied perpendicular to the electrodes, making an angle $\theta = 14°$ with the CNT. **b** Illustration of the device showing the NW with small level spacing contacted to a normal reservoir (green) and to a superconducting reservoir (brown). The gate electrode (blue) is capacitively coupled to the NW. **c** Magnetic field dependence of the superconducting gap $\Delta_j^*$ of our Nb/Pd contacts using an Al/AlO$_x$/Pd/Nb junction.

superconducting contact (S). Two other Nb/Pd outer contacts went opaque at low temperature and could not be used for transport. They were left open-circuited during the whole experiment and thus simply acted as local scatterers, as will be discussed below. A voltage $V_g$ is applied to an Al/AlO$_x$ bottom gate to modulate the chemical potential in the wire. The CNT makes an angle of 14° with the external magnetic field $B$. The diameter of the CNT, $D = 1.4$ nm, was measured a posteriori with Raman spectroscopy. It gives an orbital g-factor of $g_{orb} = ev_F D / 4\mu_B \approx 4.8$ which is used for all analysis and simulations in the following, with $\mu_B$ the Bohr magneton[12]. The whole experiment is performed at a base temperature of 36 mK. We have characterized independently the BCS superconducting correlations of our Nb/Pd electrodes as a function of magnetic field by measuring Al/AlO$_x$/Pd/Nb tunnel junctions (see Methods and Supplementary Discussion 1). We found $\Delta_j^* = 0.265$ meV, an in-plane critical field $B_{//}^c \approx 6.5$ T and an out-of-plane critical field $B_\perp^c \approx 3.5$ T (Fig. 1c). Such a large $B_{//}^c$ is crucial for performing the magneto-spectroscopy of the NW without destroying superconductivity.

We start by characterizing our system with transport measurements at zero external magnetic field. Figure 2a shows $G_{diff}$ as a function of $V_{sd}$ and $V_g$. We observe Coulomb diamonds over 60 consecutive charge states, indicating that our CNT is indeed ultra-clean (note that part of this data map was published in ref. 21 to discuss the compatibility of the CNT stapling technique with superconducting contacts). A conductance gap centered on zero bias fluctuates with $V_g$ with a maximum observed value $\Delta^* \approx 0.32$ meV. Looking more closely at an open diamond where $\Delta^*$ is large, around $V_g \approx -670$ mV (orange box), shows the two parts of the diamond to be shifted in $V_g$ (Fig. 2b). This is characteristic of a S-NW-N junction due to the alignment of the Fermi energy in the normal contact with the BCS gap in the superconducting contact[24]. The fact that the upper diamond part is shifted to positive gate voltage compared to the lower diamond part indicates that the drain is the contact with superconducting correlations as depicted in Fig. 1. Joining the summits of the two diamond parts, there is a faint transport resonance due to a finite residual DOS at zero energy in the S contact. A $V_g$ cut at zero bias allows us to observe this resonance, as shown in Fig. 2c, which will reveal crucial to precisely track the shift of a given diamond when tuning the magnetic field. At $V_g \approx -270$ mV (red box), where $\Delta^*$ is small, an almost closed diamond is observed with a similar charge degeneracy conductance resonance at zero bias (Fig. 2b, c). The transport map at $B = 5$ T on the same range of charge states is shown in Fig. 2h. We observe again Coulomb diamonds with a gap around zero energy, compatible with a critical field $B_{//}^c = 6.5$ T. The fluctuations of the gap with charge state are however qualitatively different than at $B = 0$ T.

We understand this behavior as a mini-gap $\Delta_g$ arising from weak disorder in the NW. First of all, this is not incompatible with having an ultra-clean CNT. The CNT being ultra-clean implies low intrinsic disorder and large mean free path for the electrons. This is confirmed by a level spacing corresponding to a longitudinal confinement length of about 4.9 μm as will be detailed below. The multiple gates beneath the CNT, which are either left open-circuited or floating, can act as extrinsic scatterers by locally modifying the potential landscape and thus can induce weak disorder. Finally this is also compatible with the superconducting correlation length in the proximitized conductor which scales as $h v_F / 2\Delta^* \approx 5$ μm, and therefore would extend over the whole device. The mini-gap is defined by the energy of the lowest Andreev bound state (ABS)[25]. Disorder in coherent conductors is expected to lead to mesoscopic fluctuations of physical parameters of the system, following universal probability distributions that can be inferred from random matrix theory (RMT)[26,27]. In particular the level spacing $\delta$ of a disordered quantum dot should follow a universal Wigner surmise (WS) distribution, as was recently observed[28]. In the case where superconductivity is proximity induced in the normal

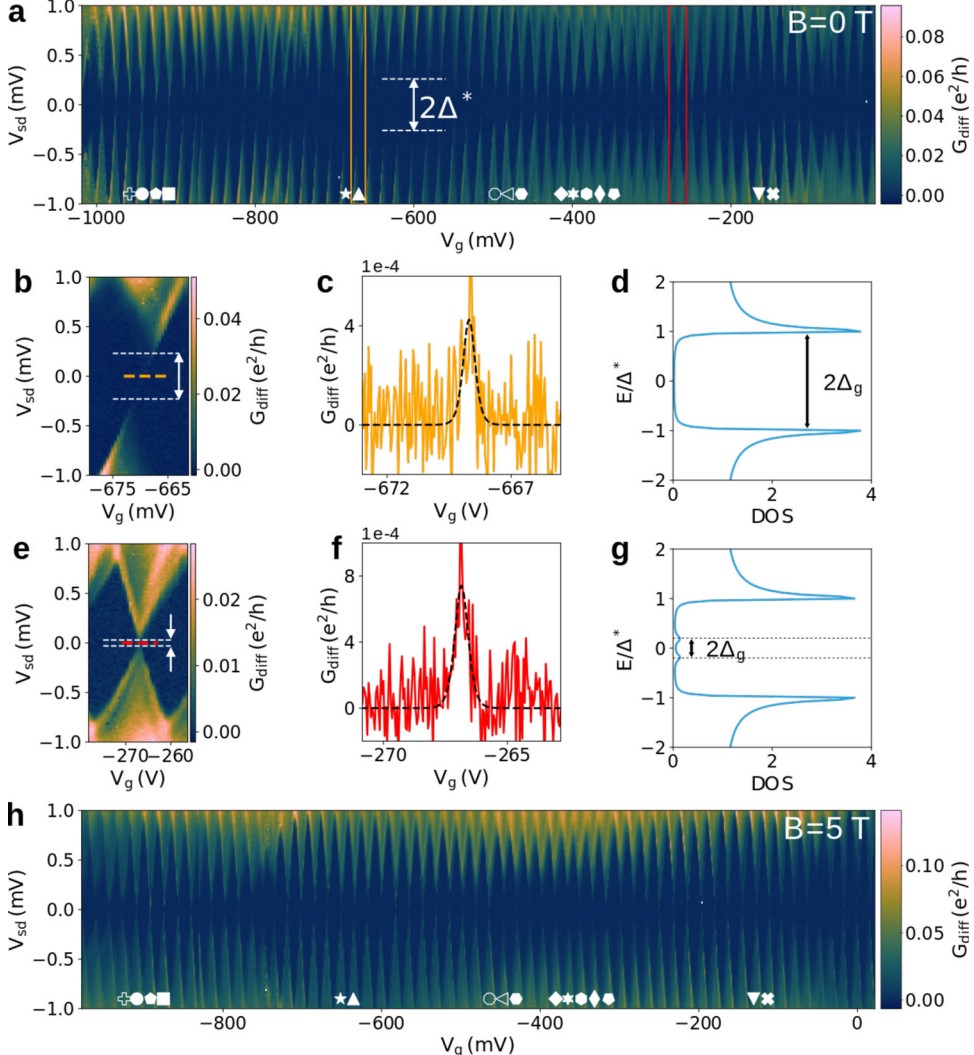

**Fig. 2 | Transport through a S-Nanowire-N system. a** Differential conductance $G_{\mathrm{diff}}$ as a function of bias voltage $V_{\mathrm{sd}}$ and gate voltage $V_g$ at $B = 0$ T. Zoom on an opened (**b**) and closed (**e**) Coulomb diamond indicated respectively by the orange and red rectangles in **a**. The corresponding differential conductance $G_{\mathrm{diff}}$ at zero bias along the degeneracy line indicated by orange and red dashed lines in **b** and **e** are shown respectively in **c** and **f** as plain curves. The black dashed line in each panel is a fit considering sequential tunneling. The minigap gap in Coulomb diamonds is indicated by white dashed line and arrow in **b** and **e** and is respectively illustrated in **d** and **g** showing a BCS DOS with Andreev state defining the position of the minigap. **h** Differential conductance $G_{\mathrm{diff}}$ as a function of bias voltage $V_{\mathrm{sd}}$ and gate voltage $V_g$ at $B = 5$ T.

conductor, mesoscopic fluctuations of the mini-gap were predicted to follow the universal Tracy-Widom (TW) distribution[25], a feature never studied nor observed experimentally so far. RMT predicts different distributions depending if time reversal symmetry is present (zero magnetic field) or broken (finite magnetic field). At zero magnetic field, the distributions should belong to the Gaussian orthogonal ensemble (GOE). At finite magnetic field, the distributions should be in the Gaussian unitary ensemble (GUE). We construct the histograms of $\Delta_g$ in our system at $B = 0$ T and $B = 5$ T (Fig. 3) in order to confront them to the predictions of RMT (Methods section). First of all, we observe a qualitative change in the distribution of $\Delta_g$ from $B = 0$ T to $B = 5$ T as it becomes narrower with increased maximum. The natural variable of the WS distribution is the scaled variable $\Delta_g/\langle\Delta_g\rangle$ with $\langle\dots\rangle$ denoting ensemble averaging. The only parameter, which encodes the universal behavior of the WS distribution, is therefore $\langle\Delta_g\rangle$. This means that for a given Gaussian ensemble, the mean value of the distribution $\langle\Delta_g\rangle$ controls the width and concomitantly the height of the distribution through its normalization. Similarly for a given $\langle\Delta_g\rangle$, the width and height of the distribution are controlled by the choice of the Gaussian ensemble, with the distribution in GUE being narrower and higher than

the distribution in GOE. It goes similarly for the TW distribution. This property of the universal distributions we are considering allow us to qualitatively distinguish between models before considering more quantitative analysis, without any fitting parameter. We show in Fig. 3 the WS distributions in GOE or GUE, using $\langle\Delta_g\rangle = 0.112$ meV at $B = 0$ T and 0.088 meV at $B = 5$ T, extracted from the raw data. We observe a good agreement between the data at zero field and the WS distribution in GOE rather than in GUE. Conversely at finite field we observe a good agreement between the data and the WS distribution in GUE rather than in GOE. This is in line with the RMT predictions. The analysis with the universal TW distribution also shows a good qualitative agreement, with a clear transition from GOE at zero field to GUE at finite field. In the case of TW distribution, it is noteworthy that we do not find a physical solution for GUE at zero magnetic field and that only the GUE distribution at finite magnetic field is physically consistent (see Methods). This qualitative analysis is fully confirmed by a quantitative model selection analysis based on comparing the Bayesian information criterion of each model for the two data sets (see Supplementary Discussion 2). Overall this is a strong indication that the fluctuations of $\Delta_g$ in our NW follow a universal distribution.

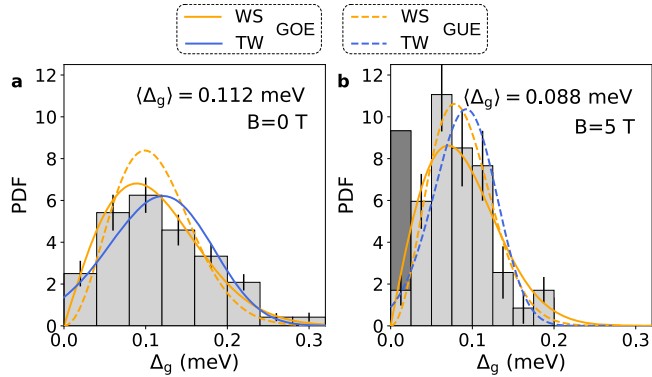

**Fig. 3 | Mini-gap distributions.** Normalized histograms of the mini-gap $\Delta_g$ for $B = 0$ T (**a**) and $B = 5$ T (**b**). The bar height is the value of the probability density function (PDF) at the bin, normalized such that the integral over the range is 1. Curves are PDF of Wigner surmise (WS) in orange and Tracy-Widom (TW) in blue, with solid lines for Gaussian orthogonal ensemble (GOE) and dashed lines for Gaussian unitary ensemble (GUE). The dark gray bar at $\Delta_g$ in **b** indicates the weight of $\Delta_g$ values that could not be evaluated.

Disorder induced mini-gap is predicted to lead to clustering of nontopological end states near zero energy at finite magnetic field[29]. To investigate this phenomenology we now perform magneto-spectroscopy of our device. As the charge ground state shifts with magnetic field $B$, the position of the Coulomb diamond needs to be tracked in order to properly fix the energy reference. This is done by measuring the position of the charge degeneracy resonance as shown in Fig. 2b, e for each value of $B$. Doing so for several consecutive charge states will allow us to reconstruct the addition spectrum of our CNT nanowire (see Supplementary Discussion 3) which we now discuss first, as it will give us more insights on the energy scales in our system. For a CNT, the canonical magneto-spectrum of a single shell, or orbital, made of four states combining spin ($\uparrow$, $\downarrow$) and valley ($K$, $K'$) degrees of freedom is shown in Fig. 4a. Each state disperse in magnetic field with a slope ($\pm 1/2 g_s \pm g_{orb}$)$\mu_B$ with $g_s = 2$ the electron g-factor and $\mu_B$ the Bohr magneton. The zero field splitting between Kramers pairs is $\sqrt{\Delta_{SO}^2 + \Delta_{KK'}^2}$, with $\Delta_{SO}$ the SOI amplitude and $\Delta_{KK'}$ the inter-valley coupling[12]. Two shells are separated in energy by the level spacing $\delta$ (Fig. 4b). The levels from two orbitals will inevitably cross, which happens first for ($g_s + 2 g_{orb}$)$\mu_B B \approx \delta$. As $\delta$ decreases so does the magnetic field at which the first crossing occurs. Each charge ground state then follows a more complicated dispersion, reminiscent of a Fock-Darwin spectrum (Fig. 4c)[30]. We show in Fig. 4d eight measured states spanning 3 orbitals that we label $M$, $N$ and $O$. They are aligned by adjusting multiple anti-crossings as hinted by black dashed lines. State labeled $N_2$ is missing as the corresponding Coulomb peak resonant line could not be resolved. The resulting low magnetic field spectrum shows shells of four states with lifted degeneracy, as expected for a CNT, with $\sqrt{\Delta_{SO}^2 + \Delta_{KK'}^2} \approx 100\,\mu$eV and $\delta \approx 325\,\mu$eV, matching remarkably the full length $L \approx 5\,\mu$m of our CNT.

Magneto-spectroscopy is then performed by fixing for each value of $B$ the value of $V_g$ at the charge degeneracy resonance before doing a bias trace. The resulting $B - V_{sd}$ map of a charge ground state, labeled by a white filled diamond in Fig. 2a, is shown in Fig. 5a as a typical example. We observe a complex subgap spectrum, symmetric in $B$, with states that disperse smoothly towards zero energy as indicated by white arrows. They remain pinned at zero energy over a significant magnetic field range of about 800 mT, centered on $\pm 1.8$ T and $\pm 3.5$ T ($V_{sd}$ cuts shown in Fig. 5c, d), although they cannot be topological in our system. We find such zero energy pinned states for multiple charge ground states, as indicated by filled symbols in Fig. 2a with all the

corresponding magneto-spectra shown in the supplementary discussion 6. The ubiquity of this behavior is expected for nontopological zero energy modes originating from mini-gap clustering[29] in contrast to other mechanisms that lead to zero energy modes only for particular set of parameters.

In addition to Andreev resonances, quasiparticle peaks are also present in the subgap conductance due to the finite DOS in the SC contact. They present sharper and more complicated variations. We confirm that they are the excited states of the NW by showing the calculated first three excited states as dashed blue, red and yellow lines respectively on Fig. 5a. This excitation spectrum is calculated from the parameters of the measured addition spectrum with the SOI value $\Delta_{SO} = -40\,\mu$eV as the only fitting parameter (see Supplementary Discussion 4). It is noteworthy that such a simple model reproduces qualitatively many of the excited states evolution with magnetic field. In particular, the model captures the overall subgap states position below 1.7 T, above 3.5 T including the diamond shape around 4 T as well as the apparent sticking to zero energy around 1.7 T and 3.5 T. This further confirms both the weak SOI and the 1D character as several states lie within $\Delta^*$ (indicated by white dashed lines). This is further validated by the magneto-spectroscopy of the previous and next charge ground states for which the same calculated excitation spectrum also matches the subgap quasiparticle peaks (see Supplementary Discussion 5). At every kink of a state dispersion, the orbital configuration of the given state is modified. It can explain the observed sudden change of conductance peaks amplitudes as the coupling to the leads depends on the state wave function. Interestingly, we observe that the excited states present several "zigzags" around zero energy, which is due to the NW states being close to each other and crossing several times over a given magnetic field range. Where it happens the amplitude of the ABS peak is enhanced, as visible around $\pm 1.7$ T and $\pm 3.5$ T (Fig. 5b), because ABS also provide DOS for the normal state to penetrate within the gap. This is particularly visible near $\pm 1.9$ T where the splitting of the first excited state (in blue) away from zero energy leads to a decrease in the amplitude of the ABS peak that remains at zero energy until $\pm 2.3$ T. Such "zigzags" or oscillations near zero energy were proposed to explain how nontopological ABS can apparently stick to zero energy in a NW with inter-subband coupling[10,11]. In our device, the intra-band coupling of the K and K' valleys only occurs at $B \cdot 0.2$ T so that we are effectively in the single transverse subband limit. Here we find that even in this limit, the condition that $\delta$ is small compared to $2g\mu_B B$ at a given $B$ (a condition met for 1D NW) also leads to nontopological states apparently sticking to zero energy.

Our study shows that a nontopological ZBCP phenomenology mimicking the sought-after Majorana modes can appear even in elemental NW systems with weak SOI. It is compatible with a mechanism based on mesoscopic fluctuations of the induced superconducting gap that we evidenced in our device. In addition, we achieved the 1D limit and observed that the resulting spectrum leads to other subgap features that can also mimic pinned zero energy states. This confirms that conductance measurements alone cannot provide conclusive observations of TS in 1D systems. In addition, even cross-correlation transport measurements at both ends of a nanowire would also not be conclusive on their own as mesoscopic fluctuations of the proximity induced superconducting gap leads to zero energy states clustering at both edges of the 1D conductor[29]. The observation of a ZBCP in transport measurement is a key signature of TS but it will need to be complemented by other signatures. Up to now, other transport signatures of TS such as, for example, the gap closing and reopening at the topological phase transition were not observed. There are promising alternative probes for providing key signatures of TS, such as microwave cavity photons for example[31,32], which in combination with transport measurement, could prove more discriminant.

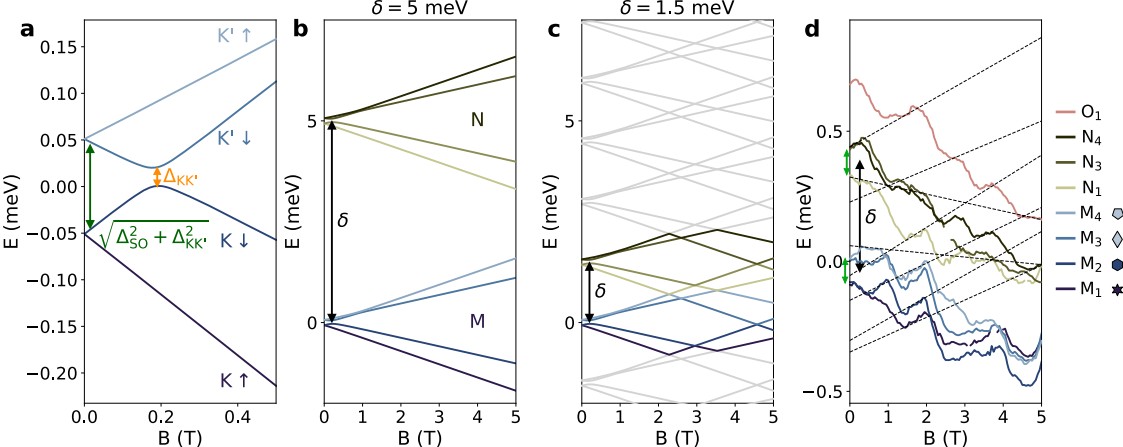

**Fig. 4 | Spectrum of a CNT nanowire. a** Magneto-spectrum of a single shell of a CNT at small magnetic fields showing the dispersion of the four states $\{K\uparrow, K\downarrow, K'\uparrow, K'\downarrow\}$. The green arrow indicates the zero field splitting between Kramers pairs $\sqrt{\Delta_{SO}^2 + \Delta_{KK'}^2}$ and the orange arrow indicates the valley coupling $\Delta_{KK'}$. **b** Calculated addition spectrum of a CNT for two consecutive shells labeled M (blue) and N (brown), as a function of magnetic field $B$. The parameters, as defined in the main text, are $\Delta_{SO} = -40\,\mu eV$, $\Delta_{KK'} = 87\,\mu eV$, $g_{orb} = 4.8$, $\theta = 14°$ and $\delta = 5\,meV$.

**c** Same spectrum as in **a** but with $\delta = 2\,meV$. The gray lines correspond to states of other shells below and above M and N. **d** Eight experimental energy levels dispersion of our CNT device as a function of magnetic field, extracted by following the position of Coulomb diamonds degeneracy point as measured in Fig. 2b, e. The black dashed lines are guides to the eye showing the continuity of an energy levels through the different ground states. In the legend, the symbols corresponds to the symbols of Fig. 2a.

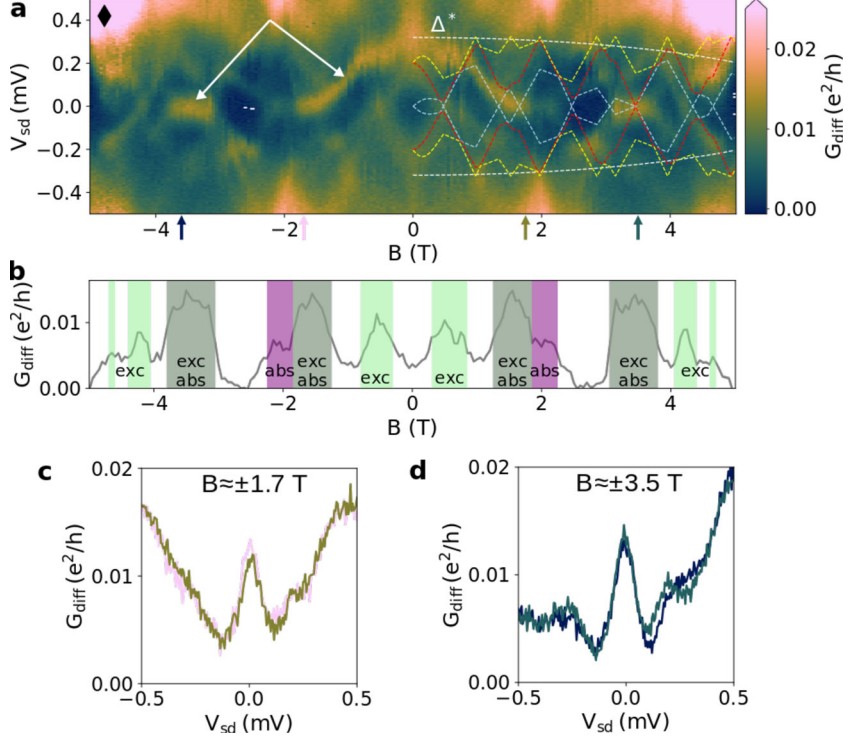

**Fig. 5 | Magneto-spectroscopy of a long nanowire with weak SOI. a** Differential conductance $G_{diff}$ as a function of bias voltage $V_{sd}$ and magnetic field $B$ at the Coulomb diamond degeneracy point indicated by a diamond symbol around $V_g \approx -365\,mV$ in Fig. 2a. The white dashed line indicates the position of $\Delta^*(B)$.

The energy of the first 3 excited states are shown as blue, red and yellow dashed lines. **b** Cut at zero bias $V_{sd} = 0$ of **a** with highlighting of the contribution to each ZBCP as excited states ("exc", green), ABS ("abs", purple) or both (gray). **c, d** $V_{sd}$ cuts at $B \approx \pm 1.7\,T$ and $B \approx \pm 3.5\,T$ respectively as indicated by arrows in **a**.

## Methods

### Fabrication of Al/AlO$_x$/Pd/Nb junctions

The Al/AlO$_x$/Pd/Nb control junctions are fabricated with the following steps. First Au(30 nm) contacts are deposited. Then PMMA walls of 450 nm are lithographied to fabricate the junction. An O$_2$ RIE plasma is done for 1 min before pumping overnight the sample in the load-lock

of the thin film evaporator at a pressure of ~$2\times10^{-8}$ mbar. Ti/Al(3/30 nm) is evaporated under an angle of 30°. Oxidation is done in the evaporator load-lock chamber with an O$_2$ pressure of 0.58 mbar for 4 min. The load-lock chamber is pumped back to ~$2\times10^{-8}$ mbar in 25 min. Pd(10 nm) is evaporated under an angle of 30° and then Nb(80 nm) is evaporated under zero angle.

## Minigap distributions

The number of bins $N_{bins}$ of the histograms of the minigap $\Delta_g$ value shown in Fig. 3 of the main text is chosen using the conventional rule that it should be the root mean square of the number of samples rounded to the ceiling integer, which yields $N_{bins} = 8$. The error bars of the histograms are estimated as follows : we estimate the uncertainty on the minigap value to be of the order of $10\,\mu eV$. We resample $n = 2000$ times the set of minigap values assuming a normal distribution with standard deviation of $10\,\mu eV$ and calculate for each set the corresponding histogram, keeping the bins constant. The standard deviation of each bar of the histogram is calculated from the set of all histograms and used as error bar.

The histograms of $\Delta_g$ are confronted to distributions arising from RMT. The Wigner surmise (WS) probability density function (PDF) is

$$P_{WS}(s, \beta) = a_\beta s^\beta e^{-b_\beta s^2}, \tag{1}$$

with

$$a_\beta = 2 \frac{\left[\Gamma\left(\frac{\beta+2}{2}\right)\right]^{\beta+1}}{\left[\Gamma\left(\frac{\beta+1}{2}\right)\right]^{\beta+2}} \quad \text{and} \quad b_\beta = \frac{\left[\Gamma\left(\frac{\beta+2}{2}\right)\right]^2}{\left[\Gamma\left(\frac{\beta+1}{2}\right)\right]^2}, \tag{2}$$

with $\Gamma(z) = \int_0^\infty x^{z-1}e^{-x}dx$ the Gamma function and $\beta$ the Wigner-Dyson index. The Gaussian orthogonal ensemble (GOE) corresponds to $\beta = 1$ and the Gaussian unitary ensemble (GUE) to $\beta = 2$. In our situation, the WS distribution variable $s$ is $\Delta_g$ in units of the minigap mean value $\langle\Delta_g\rangle$, with $\langle\dots\rangle$ denoting the ensemble averaging. It follows that $\sqrt{var[P_{WS}(s, \beta=1)]} = \langle\Delta_g\rangle\sqrt{\frac{4}{\pi} - 1}$ and $\sqrt{var[P_{WS}(s, \beta=2)]} = \langle\Delta_g\rangle\sqrt{\frac{3\pi}{8} - 1}$, with $var[x] = \langle x^2\rangle - \langle x\rangle^2$ the variance. Therefore $\langle\Delta_g\rangle$ is the only parameter that controls the distribution.

The Tracy-Widom (TW) probability density functions $P_{TW}(-s', \beta)$ with $\beta = 1$ for GOE and $\beta = 2$ for GUE are calculated using the Edelman and Persson algorithm[33]. According to RMT, the TW distribution variable is $-s' = (\Delta_g - E_g)/w_g$ with $E_g$ and $w_g$ respectively the gap and the width of the gap distribution in the mean-field approximation[25]. For TW distributions, there are therefore two parameters that can be evaluated from the data directly. We have the following relations

$$\langle\Delta_g\rangle = -w_g\langle P_{TW}(x, \beta)\rangle + E_g \tag{3}$$

$$\sqrt{var[\Delta_g]} = w_g\sqrt{var[P_{TW}(x, \beta)]}, \tag{4}$$

where $x$ is the natural (not scaled) variable of TW distribution. The corresponding values are

$$\langle P_{TW}(x, \beta=1)\rangle = -1.20653, \quad \sqrt{var[P_{TW}(x, \beta=1)]} = 1.26798 \tag{5}$$
$$\langle P_{TW}(x, \beta=2)\rangle = -1.77109, \quad \sqrt{var[P_{TW}(x, \beta=2)]} = 0.90177$$

We can therefore estimate $E_g$ and $w_g$ from $\langle\Delta_g\rangle$ and $\sqrt{var(\Delta_g)}$ extracted in the raw data, assuming a value for $\beta$. We have

$$w_g(B, \beta) = \frac{\sqrt{var[\Delta_g]_B}}{\sqrt{var[P_{TW}(x, \beta)]}} \tag{6}$$

$$E_g(B, \beta) = \left[\langle\Delta_g\rangle_B + w_g(B, \beta)\langle P_{TW}(x, \beta)\rangle\right]^{-1} \tag{7}$$

The experimental values extracted from the datasets are $\langle\Delta_g\rangle_{B=0T} = 0.112$ meV, $\langle\Delta_g\rangle_{B=5T} = 0.088$ meV, $\sqrt{var[\Delta_g]_{B=0T}} = 0.065$ meV and

$\sqrt{var[\Delta_g]_{B=5T}} = 0.039$ meV. From this we find the following values of $w_g(B, \beta)$ and $E_g(B, \beta)$:

| | |
|---|---|
| $w_g(B=0\,T, \beta=1) = 0.052\,meV$ | $E_g(B=0\,T, \beta=1) = 0.050\,meV$ |
| $w_g(B=0\,T, \beta=2) = 0.072\,meV$ | $E_g(B=0\,T, \beta=2) = -0.017\,meV$ |
| $w_g(B=5\,T, \beta=1) = 0.031\,meV$ | $E_g(B=5\,T, \beta=1) = 0.051\,meV$ |
| $w_g(B=5\,T, \beta=2) = 0.043\,meV$ | $E_g(B=5\,T, \beta=2) = 0.012\,meV$ |

First we see that there is no physical solution at $B = 0$ T in the GUE ($\beta = 2$) as it gives a negative mean-field gap value. Only the TW distribution in GOE ($\beta = 1$) can describe the data. Following this, we see that only the TW distribution in GUE ($\beta = 2$) can consistently describe the data at finite field $B = 5$ T. Indeed we find that the mean-field gap at $B = 5$ T in GOE ($\beta = 1$) would be the same than the mean-field gap at zero field, which is not expected. Instead we have $E_g(B = 5T, \beta = 2) < E_g(B = 0T, \beta = 1)$ as expected. Therefore we find that the TW distribution not only agrees well with our data distribution, but also that it shows a transition from GOE to GUE from zero magnetic field to finite magnetic field as expected from RMT.

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

## Acknowledgements
This research was supported by the EMERGENCES grant MIGHTY of ville de Paris (MRD) and the QuantERA project SuperTop (TK).

## Author contributions
M.R.D. conceived and supervised the project. L.C.C fabricated the device. L.C.C., L.J., and M.R.D. performed the measurements. L.C.C., L.J., W.L., A.C., T.K., and M.R.D. analyzed the data. L.J. fabricated, performed the measurements and the analysis of the tunnel junction control device. L.C.C., L.J., W.L., A.C., T.K., and M.R.D. discussed the data, contributed to their interpretation and wrote the manuscript.

## Competing interests
The authors declare no competing interests.
