## [Peer Review File · Nature Communications]

Zero energy states clustering in an elemental nanowire coupled to a superconductorREVIEWER COMMENTS

Reviewer #1 (Remarks to the Author):

In this work, the authors studied a nanotube quantum dot coupled to a superconductor and a normal metal, and presented a very interesting and important topic on the observation of zero energy states, which was attributed to clustering of trivial states. Since a zero-bias conductance peak (ZBCP) was assigned as a signature of Majorana zero mode, it is crucial to observe ZBCPs and meanwhile to clarify the origins, either trivial or non-trivial. This work showed a possible trivial mechanism in mesoscopic superconducting hybrids. However, I raised several concerns during the review which need to address.

1. The authors emphasized that the carbon nanotube was ultra clean, resulted from the stapling. However, they also claimed weak disorder, required for the gap distribution and the zero energy states. Was it clean or disordered? This point should be clarified in detail.
2. Lines 33-39: The authors mentioned "In the case of TS...", but they did not expect TS in the system at all. In addition, there was definitely a quantum dot in the CNT, so the device was in the 0D regime. Why the authors call it 1D regime? It sounds misleading to me.
3. The device. There were four contacts and two gates, and the authors used two contacts and one gate. They stated that the green one worked as a normal reservoir and the brown one as a superconducting reservoir. How did the authors determine such behavior? How soft/hard was the superconducting gap for each contact? (Line 46, "a large residual density of states (DOS) at low energy...". Line 72, "a finite residual DOS at zero energy in the S contact".) Since the 5 μm long CNT formed a single quantum dot, how about the other two contacts? What was their effect? Detailed explanation and extra data are required for the readers.
4. The contact itself. In Fig. S1, the authors showed the tunnel measurements on Al/AlOx/Pd/Nb junctions at 300 mK, which was low enough for Al and Pd/Nb. Since Al and Pd/Nb are both superconducting at B=0, the tunnel measurement should show a hard superconducting gap with a size of their addition. However, the data showed only a soft gap and noisy curves. Why? At finite B, Al is normal and the measured gap should reflect Pd/Nb only. I could not find the jump of the gap size from B=0 to B=0.5 T. Why? In addition, how did the authors extract Fig. 1c? Was it reliable? Please show the detailed analysis.
5. Superconducting proximity in the CNT. Since I can only see a very soft gap in the contact itself, I doubt the quality of the superconducting gap in the CNT. A relatively hard superconducting gap was the footstone. The authors cited Ref. 24 as an important reference, which treated a quantum dot coupled to a superconductor. However, the CNT had a length of 5 μm in the current work. Which part of the CNT was superconducting? Which part did the conductance measurement probe? Where did the zero energy states and the subgap states locate? Please add a detailed explanation.
6. Line 58. $\Delta^*=0.265$ meV. How did the authors extract the induced gap in the CNT? It seems to be the gap of the contact itself. If so, why it was much smaller than the gap of Nb?

7. Please find a comparison between Fig. 2a in the current manuscript and Fig. 3 in Ref. 20 below. Was part of the data in Fig.2 published in Ref. 20? If so, did the authors clarify this point at the submission of the manuscript? Since Figs. 2b, 2c, 2e, 2f and 3a were extracted from Fig. 2a, extra cautions should be added. Could the authors clarify this point?

8. Line 106. “We observe a remarkable agreement...”. Please note that there were large error bars in Fig. 3. How remarkable was it? It is crucial to determine which method was the best. Please add a quantitative and detailed comparison between different methods for both $B=0$ and $B=5$ T.

9. Line 162. “It remarkably reproduces most of the excited states evolution ...”. I am sorry to say that I could not find the “remarkably”. Without the dashed lines, I can only see randomly distributed weak subgap states. Since a main point of the current work is the zero energy states, please validate the statement.

10. Fig. 4d. If I am correct, Fig. 4d was extracted from a measurement by fixing B values and sweeping the gate voltage, and a 2D conductance map was obtained. Right? If not, could the authors explain the measurement? In any case, to have a better understanding of the data and the statement of “remarkably” in Line 140, could the authors add the measured data?

Reviewer #2 (Remarks to the Author):

Contamin et. al. performed magneto transport measurements on an ultra clean, small diameter carbon nanotube device with normal and superconducting leads along with a side plunger gate to tune the levels of the resulting quantum dot. They perform bias spectroscopy and analyze detailed sub-gap features to infer universal RMT type behavior, and more specifically they observe somewhat robust zero bias conductance peaks (ZBCPs).

The stated value of this work is that it demonstrates the observation of ZBCPs in a truly 1D electronic system proximitized to a superconductor despite the absence significant spin orbit coupling. In showing this behavior, this work adds to a body of work showing that ZBCPs aren't decisive evidence for Majorana bound states.

To this reviewer, this is indeed valuable, and the work warrants publication with some revision. The data collection and analysis is thorough and one need not look much further than Figure 5 and the supplementary figures to see how prevalent ZBCPs are in this system.

Regarding changes this reviewer sees as important:

(1) The data and discussion for Figure 3 seems to need a bit more analysis/consideration (perhaps in the supplement). The statement that the data clearly goes from GOE to GUE at low to high field doesn't seem to be fully evident in the data itself. These are presumably normalized curves, and so the apparent height of the peaks seem highly contingent on the high Δg tails. Reworking this discussion or providing more statistical support would make the assertions more believable.

(2) The data is clear and the analysis reasonable, but to this reviewer, the strength of the conclusions appear to overreach a bit. The model fits in Fig. 5a appear to mostly agree with the observed phenomena, but not completely, so it is not obvious that the work decisively showed that the observations were purely from the NW spectrum. A more modest phrasing might be more warranted. More to the point, statements like "This definitely shows that conductance measurements cannot provide conclusive observations of TS in 1D systems" appear to weigh in to a debate about Majorana zero modes without sufficient precision in the language. Just because this manuscript showed analogous features that the community had taken as signatures of Majorana physics in a different system doesn't rule out that further analysis and theory *could* identify a decisive feature of TS in transport measurements. To this reviewer, the data does well enough to drive home the point that ZBCPs aren't decisive evidence, and it would serve this work better to avoid controversial/imprecise language.

We would like to sincerely thank both reviewers for the time they spent with our manuscript, their insightful comments and suggestions. Both reviewers express a very positive general opinion on our work (“very interesting and important topic”, “valuable”) and essentially recommend publication given we address their remarks or concerns.

The main changes in the main text are:

- The discussion in the main text on the gap fluctuation and agreement with universal distribution has been modified and extended to reflect the qualitative and quantitative analysis.
- The conclusion has been amended.
- Several changes throughout the text to reflect reviewers comments.

The main changes in the supplementary material are:

- Discussion 1 on the superconducting gap measurement in control Al/AlO_x/Pd/Nb junctions has been extended.
- Discussion 2 on the model selection for confronting the universal distributions to the observed gap fluctuations histogram has been added.
- Discussion 3 on how the addition spectrum is extracted has been added.

We would like to answer here a comment/question that was raised by both reviewers before going into each reviewer’s remarks (it is question 8 of reviewer #1 and question 1 of reviewer #2). It concerns the agreement between the universal distributions of RMT and the experimental histograms of Δg in Fig. 3. Both reviewers ask for a more detailed and quantitative analysis given the large error bars; essentially it consist in performing model selection. We agree with the reviewers that it is an important point required to back up our claim. We thus calculated the Bayesian Information Criterion (BIC), which is one of the standard criteria used in model selection, for the various distributions at both $B=0T$ and $5T$, taking into account the error bars. This is the conventional methodology for model selection. Doing so fully confirms our claim : at $B=0T$, we find $BIC(WS-GOE) < BIC(TW-GOE) < BIC(WS-GUE)$ and at $B=5T$, we find $BIC(WS-GUE) < BIC(WS-GOE) < BIC(TW-GUE)$.

The conclusion of this model selection analysis is as follow:

1. The Wigner surmise distributions (in the correct Gaussian ensemble) quantitatively account better for the data than the Tracy-Widom distribution (as was qualitatively observed).
2. The model selection methodology based on BIC quantitatively assesses that the preferred model for the data at $0T$ is the WS distribution in GOE and that the preferred model for the data at $5T$ is the WS distribution in GUE, confirming the expected crossover due to time reversal symmetry breaking.

Therefore, the qualitative analysis we provided in the initial submission is fully validated by this quantitative analysis. However we agree that our initial statement “We observe a remarkable agreement between the WS distribution in GOE at zero field and in GUE at finite field as expected by RMT, without any fitting procedure.” might be misleading because it might not be so easy to qualitatively infer from the curves. We therefore amended the text to reflect that a quantitative model selection procedure confirms predictions from RMT.

The discussion on the model selection has been added in the supplementary material. The discussion in the main text on the distributions has been modified and extended.

Response to reviewer #1

Reviewer #1 (Remarks to the Author):

In this work, the authors studied a nanotube quantum dot coupled to a superconductor and a normal metal, and presented a very interesting and important topic on the observation of zero energy states, which was attributed to clustering of trivial states. Since a zero-bias conductance peak (ZBCP) was assigned as a signature of Majorana zero mode, it is crucial to observe ZBCPs and meanwhile to clarify the origins, either trivial or non-trivial. This work showed a possible trivial mechanism in mesoscopic superconducting hybrids. However, I raised several concerns during the review which need to address.

We would like to thank the reviewer for his/her positive assessment. We reply point by point below to the concerns and remarks of the reviewer.

1. The authors emphasized that the carbon nanotube was ultra clean, resulted from the stapling. However, they also claimed weak disorder, required for the gap distribution and the zero energy states. Was it clean or disordered? This point should be clarified in detail.

The reviewer is right that this point needs to be made clearer. The carbon nanotube itself is ultra-clean as provided by the stapling technique. It implies that the mean free path of electrons in the CNT is large due to low intrinsic disorder. This point is confirmed by the level spacing δ that we extract corresponding to a system length of 4.9 μm . Weak disorder can arise from the gates beneath the CNT which can act as scatterers because they can for example induce local modulation of the potential landscape. We have added a sentence in the main text to explain this point.

2. Lines 33-39: The authors mentioned "In the case of TS...", but they did not expect TS in the system at all. In addition, there was definitely a quantum dot in the CNT, so the device was in the OD regime. Why the authors call it 1D regime? It sounds misleading to me.

We indeed do not expect TS in our device however our aim is to investigate superconductivity induced in a 1D conductor in conditions required to realize TS (which requires additionally SOI and Zeeman field). The current state of the field has shown that combining all the ingredients to realize TS is more complicated than anticipated so it is important to benchmark in a simpler system. It is however crucial that it meets the criteria that are required to realize TS, otherwise it would not serve as a benchmark for more complex experiments. Therefore it is why we wanted to already fulfill the requirement that $\delta \leq \Delta_T$ with the Δ_T that can be expected.

Regarding the confinement, the reviewer is right, there is obviously a longitudinal confinement, on a length that we could estimate to be of 4.9 μm . It implies a quantum dot with a small level spacing much smaller than in usually investigated quantum dots that are typically five times smaller (level spacing five times larger). Therefore our system is in the "1D" limit regarding the superconducting gap, as explained in the main text, in contrast to previous experiments.

3. The device. There were four contacts and two gates, and the authors used two contacts and one gate. They stated that the green one worked as a normal reservoir and the brown one as a superconducting reservoir. How did the authors determine such behavior? How soft/hard was the superconducting gap for each contact? (Line 46, "a large residual density of states (DOS) at low energy...". Line 72, "a finite residual DOS at zero energy in the S contact".) Since the 5 μm long CNT formed a single quantum dot, how about the other two contacts? What was their effect? Detailed explanation and extra data are required for the readers.

We have identified the behaviour of the contacts as now explained in the main text lines 72-74. The observation of Coulomb diamond with apexes shifted in gate voltage is the signature of a S-(normal conductor)-N junction/device. As the upper parts of the diamonds are shifted to the

right (more positive gate voltage), it implies that the drain contact is the one having superconducting correlations. This fact is related to our single step fabrication of Nb/Pd contacts. We believe that one of the two contacts (the source here) has a lateral shift between the superconducting layer and the normal proximitized one due to a shadow effects in the metallic deposition. Hence, this can turn one of our Nb/Pd contacts into a Pd contact.

The hardness of the contact is difficult to infer in our case since we do not observe cotunneling lines inside the Coulomb diamond. Hence, the bias dependence near the shifted diamond apexes is distorted by Coulomb blockade and the hardness of the gap is difficult to deconvolve.

The other contacts were left open-circuited throughout the whole experiment so that no current could flow through them. They therefore had no impact on the transport apart from playing the role of scattering center as discussed in point 1. To add even more detail, the contact on the left of the (S) contact went open-circuit during the cooldown (typically a bad contact with Schottky barrier behaviour) and the the contact on the right of the (N) contact had a contact resistance 10 times larger at 4K and was therefore very opaque and not usable for transport measurement. We did not have the opportunity to try measuring through this contact before the device had to be warmed up, so we have no data to show.

4. The contact itself. In Fig. S1, the authors showed the tunnel measurements on Al/AlO_x/Pd/Nb junctions at 300 mK, which was low enough for Al and Pd/Nb. Since Al and Pd/Nb are both superconducting at B=0, the tunnel measurement should show a hard superconducting gap with a size of their addition. However, the data showed only a soft gap and noisy curves. Why? At finite B, Al is normal and the measured gap should reflect Pd/Nb only. I could not find the jump of the gap size from B=0 to B=0.5 T. Why? In addition, how did the authors extract Fig. 1c? Was it reliable? Please show the detailed analysis.

Small Al/AlO_x/Nb (and similarly Al/AlO_x/Pd/Nb) junctions (typically micron size) are difficult to realize due to the difficulty to realize a pinhole free AlO_x/Nb interface. Although not ideal, such a DOS measurement is enough to measure the evolution of the gap edge as a function of the magnetic field and extract the critical field, which is the only quantity we needed and exploited to independently confirm that we could still observe superconductivity in our CNT device at 5T.

The noisy curve arise from the measurement setup of the fridge we used for this measurement which is not wired for low-noise measurement. We preferred to show raw data rather than filtered data with moving average. When reworking on this, we realized that the data of Fig S1b were having a moving average with a window of 3 points, while the S1a data were raw. In the new figure, each panel is now doubled to present the raw data and data with a moving average window (size 3) to show more clearly the gap features.

We do observe a transition where the Al gap is destroyed by the magnetic field; In figure S1a, there is a jump from the trace at 0T to the trace at 0.5T and the evolution of the gap is then much smoother when increasing the field. We acknowledge it is difficult to see with the raw noisy data but is clearer with moving average. We initially did not show the data at 0T in panel b and have added it for completeness. The parameters of the BCS density of state (DOS) are extracted by fitting each curve with the BCS DOS including a Dynes parameter (equivalently decoherence term) which is used to model the soft gap. This is now explained in the supplementary text.

We have added the theory curves with which we extract the values of the gap to the panel showing traces as a function of perpendicular magnetic field with a moving average window of size 3.

5. Superconducting proximity in the CNT. Since I can only see a very soft gap in the contact itself, I doubt the quality of the superconducting gap in the CNT. A relatively hard superconducting gap was the footstone. The authors cited Ref. 24 as an important reference, which treated a quantum dot coupled to a superconductor. However, the CNT had a length of 5

μm in the current work. Which part of the CNT was superconducting? Which part did the conductance measurement probe? Where did the zero energy states and the subgap states locate? Please add a detailed explanation.

We would like to thank the reviewer for this important question. The hardness of the gap cannot alone give a hint of where superconductivity is induced. In our case, we observe fluctuations of the gap edge as shown in the conductance maps at zero and finite B-field. Such fluctuations can only be understood in our view from mesoscopic fluctuations, well accounted for by RMT. This implies that the electron trajectories which give rise to proximity effect can be as long as several times the total active length of the device. This manifests itself by the existence of very low energy subgap states as shown in the distributions shown in figure 3. This is also compatible with the superconducting correlation length in the normal proximitized conductor that scales as $h v_F / \Delta^* \approx 10 \mu\text{m}$, as v_F is large in CNT.

6. Line 58. $\Delta^ = 0.265 \text{ meV}$. How did the authors extract the induced gap in the CNT? It seems to be the gap of the contact itself. If so, why it was much smaller than the gap of Nb?*

This question is related to the above one. The shift of the apexes of the Coulomb diamonds are used to infer the S-NT-N character of our device and allow us to directly measure the gap from the distance between the apexes, as shown by the dashed lines in figure 2b and 2e. The gap of the contacts can be extracted from the Al/Alox/Pd/Nb measurements presented in FigS1. The reduced value of the Nb/Pd gap ($\sim 265 \mu\text{eV}$) with respect to the Nb bulk value 1.48 meV is due to the proximity effect in the Pd itself [Kontos et al. PRL 93, 137001, (2004)]. The small difference between the gap measured in the junction experiment and the maximum gap value observed in the CNT device ($\sim 320 \mu\text{eV}$) can arise from slightly different thickness of the Nb/Pd layers in the two devices. We have renamed the proximity induced gap in the junction Δ^*_j to avoid confusion between the two.

7. Please find a comparison between Fig. 2a in the current manuscript and Fig. 3 in Ref. 20 below. Was part of the data in Fig. 2 published in Ref. 20? If so, did the authors clarify this point at the submission of the manuscript? Since Figs. 2b, 2c, 2e, 2f and 3a were extracted from Fig. 2a, extra cautions should be added. Could the authors clarify this point?

We thank the reviewer for finding this out. We had in all honesty completely forgotten that we had already published part of Fig. 2a in Ref. 20. We therefore did not mention that point at the submission of the manuscript but it is now done. In Ref 20, we used part of Fig. 2a to only justify that the stapling technique that we developed was compatible with superconducting contacts. The in-depth analysis and investigation we perform in the current work is on a total different scale (gap fluctuations, magneto-spectroscopy) and we do not see any conflict on the scientific side here. We are discussing with the editor to see how to best mention that part of Fig. 2a was already published in ref. 20.

8. Line 106. "We observe a remarkable agreement...". Please note that there were large error bars in Fig. 3. How remarkable was it? It is crucial to determine which method was the best. Please add a quantitative and detailed comparison between different methods for both $B=0$ and $B=5 \text{ T}$.

See the general answer at the beginning of this document.

9. Line 162. "It remarkably reproduces most of the excited states evolution ...". I am sorry to say that I could not find the "remarkably". Without the dashed lines, I can only see randomly distributed weak subgap states. Since a main point of the current work is the zero energy states, please validate the statement.

We would like to mention here that we realized that our use of the word “remarkable/remarkably” was wrong because of a mistranslation from French “remarquable” which we thought more of “notable/noteworthy” (rather than outstanding). In all honesty we were surprised to see that the simple model of the calculated excited states in the CNT could reproduce so many features of the subgap states that are observed. We have reworded the sentence to reflect this more accurate wording.

We interpret the comment of the reviewer [“without the dashed lines, I can only see randomly distributed weak subgap states.”] as the fact that the presence of the dashed lines induces an interpretation bias. Well aware of that, it is why we displayed the dashed lines on one half of the data only so that raw/unbiased interpretation can still be made. We think the dashed lines are important to help the general reader (that is not used to such spectra) to understand the interpretation of the data.

Finally we would like to stress again that these dashed lines are the outcome of a simple model (with a single constrained fitting parameter), which reproduces not all but many of the observed features; to us, features like the first excited state remaining close to zero energy around $B=1.7T$ and $B=3.5T$, the diamond around $4T$ and the overall states evolution below $1.7T$ and above $3.5T$ are in our honest view well captured. However these lines only indicate the energy (or position) of the excited states with B and not the amplitude or width of the respective conductance peaks evolution with B . The peaks sometimes vanish or gets blurred in the background which makes them not as easy to identify or follow. Modeling these would have required tens of parameters and doing so would have added almost zero scientific value in our view.

10. Fig. 4d. If I am correct, Fig. 4d was extracted from a measurement by fixing B values and sweeping the gate voltage, and a 2D conductance map was obtained. Right? If not, could the authors explain the measurement? In any case, to have a better understanding of the data and the statement of “remarkably” in Line 140, could the authors add the measured data?

The reviewer is right about the way we extracted the data of Fig. 4d. We have added a full new figure in the supplementary material as well as the corresponding text to present the data acquisition and analysis to construct the spectrum of Fig. 4d.

Response to reviewer #2

Reviewer #2 (Remarks to the Author):

Contamin et al. performed magneto transport measurements on an ultra clean, small diameter carbon nanotube device with normal and superconducting leads along with a side plunger gate to tune the levels of the resulting quantum dot. They perform bias spectroscopy and analyze detailed sub-gap features to infer universal RMT type behavior, and more specifically they observe somewhat robust zero bias conductance peaks (ZBCPs).

The stated value of this work is that it demonstrates the observation of ZBCPs in a truly 1D electronic system proximitized to a superconductor despite the absence significant spin orbit coupling. In showing this behavior, this work adds to a body of work showing that ZBCPs aren't decisive evidence for Majorana bound states.

To this reviewer, this is indeed valuable, and the work warrants publication with some revision. The data collection and analysis is thorough and one need not look much further than Figure 5 and the supplementary figures to see how prevalent ZBCPs are in this system.

We would like to thank the reviewer for his/her very positive assessment.

Regarding changes this reviewer sees as important:

(1) The data and discussion for Figure 3 seems to need a bit more analysis/consideration (perhaps in the supplement). The statement that the data clearly goes from GOE to GUE at low

to high field doesn't seem to be fully evident in the data itself. These are presumably normalized curves, and so the apparent height of the peaks seem highly contingent on the high Δg tails. Reworking this discussion or providing more statistical support would make the assertions more believable.

We have answered this question at the top of the reply document where we explain how we have now added a quantitative analysis based on model selection.

In addition, we thank the reviewer for bringing his concern to our attention. First of all, the histograms are indeed normalized to confront with the probability density functions given by RMT. Second, the point is that the two distributions we confront the data to are universal. This means, as stated in the main text and explained in the methods that there is no fitting parameter. For the Wigner surmise case, the mean value of the data ensemble entirely sets the distribution which can then be computed with $\beta=1$ or 2 for GOE or GUE. As pointed out by the reviewer, the Δg tail is contingent to the maximum value of the distribution and its mean. This is because the mean of the distribution sets its width as explained in the methods. It is a clear feature of the distribution with a qualitative (and quantitative) change from GOE to GUE showing a narrowing of the distribution concomitant with an increase of its maximum value (hence the peaks heights). The same goes for the Tracy–Widom distribution.

We have reworked the discussion to make this point clearer in the main text.

*(2) The data is clear and the analysis reasonable, but to this reviewer, the strength of the conclusions appear to overreach a bit. The model fits in Fig. 5a appear to mostly agree with the observed phenomena, but not completely, so it is not obvious that the work decisively showed that the observations were purely from the NW spectrum. A more modest phrasing might be more warranted. More to the point, statements like "This definitely shows that conductance measurements cannot provide conclusive observations of TS in 1D systems" appear to weigh into a debate about Majorana zero modes without sufficient precision in the language. Just because this manuscript showed analogous features that the community had taken as signatures of Majorana physics in a different system doesn't rule out that further analysis and theory *could* identify a decisive feature of TS in transport measurements. To this reviewer, the data does well enough to drive home the point that ZBCPs aren't decisive evidence, and it would serve this work better to avoid controversial/imprecise language.*

We agree with the reviewer that a more modest phrasing is indeed more appropriate. As the reviewer points out, the data speak for themselves “that ZBCPs aren't decisive evidence” of Majorana physics. The message we wanted to deliver is that our work further strengthen the fact that observation of a ZBCP by itself is non-conclusive. Our observation of the gap universal fluctuations also indicates that even if a ZBCP is simultaneously observed at both ends of the NW, as is envisioned in cross-correlated edge measurements for identifying non-local states, it would still not be conclusive on its own. As the reviewer rightfully points out, conductance measurements could still be conclusive in the future. Actually, in our view, transport measurement of a ZBCP will still be crucial in the future to identify TS. However it will need to be complemented by other observations such as the gap closing and reopening or the coupling signatures of the zero energy mode to microwave photons as we proposed, for example.

We have amended the part of the main text accordingly to the reviewer's suggestion and have added a few sentences to better express our message.

REVIEWER COMMENTS

Reviewer #1 (Remarks to the Author):

The authors responded to all the remarks and concerns raised by both reviewers. I am pleased that many of my concerns have been addressed. However, I still have some concerns for the authors. I could not recommend the publication before addressing these issues. (The numbering below follows the last round review.)

2. The statement of the 1D limit and the word “TS”. The authors calculated a superconducting correction length of 10 μm , and the level spacing of the quantum dot corresponds to a length of 4.9 μm . Even if there is a factor of 2 to estimate the condition of the 1D regime (Line 38), it does not support the statement of the 1D limit. Regardless of the correctness of the authors’ logic, a quantum dot means a confined object in all three dimensions that a level spacing can be distinguished. With a quantum dot in the device for sure, and the authors could recognize the level spacing, thus whatever the geometry is, the statement of the 1D limit is questionable. Usually, 1D means the level spacing due to the confinement along the wire could not be distinguished, i.e., effectively a continuous sub-band. The comparison to the superconducting gap is somehow loss of fundamental. In addition, the authors kept the word “In the case of TS...” (Line 35), which is misleading. Again, they did not expect TS in the system at all, and the system was not assumed to meet the criteria of TS. So I suggest a clear statement rather than mentioning something confusing.

3. From the explanation of the authors, I noticed that one of the four superconducting contacts worked properly, providing a superconducting proximity effect to the nanotube. One of them is poorer, and could be used as a normal probe. The other two could not be used for transport measurement. Please specify the information for the readers, since the yield and reliability of this method is important, for the so-called ultra-clean nanotube.

7. Data already published in ref. 20. I found it hard to accept the reason stated by the authors, since 4 out of the 6 authors in the current work were co-authored in ref. 20. It is now clear that the authors did not specify that part of the data in the current work has been published.

Reviewer #2 (Remarks to the Author):

The authors satisfactorily addressed this reviewer's comments and thus it is suitable for publication in Nature Communications.

We would like to thank again the reviewers for their time and work. Reviewer #2 is satisfied with our first reply and recommends publication. Reviewer #1 is also pleased by our reply but still has a few concerns. We address his points below with corresponding changes in the main text, which once again improve the quality of the paper in our view.

Reviewer #1 (Remarks to the Author):

The authors responded to all the remarks and concerns raised by both reviewers. I am pleased that many of my concerns have been addressed. However, I still have some concerns for the authors. I could not recommend the publication before addressing these issues. (The numbering below follows the last round review.)

2. The statement of the 1D limit and the word “TS”. The authors calculated a superconducting correction length of 10 μm , and the level spacing of the quantum dot corresponds to a length of 4.9 μm . Even if there is a factor of 2 to estimate the condition of the 1D regime (Line 38), it does not support the statement of the 1D limit. Regardless of the correctness of the authors’ logic, a quantum dot means a confined object in all three dimensions that a level spacing can be distinguished. With a quantum dot in the device for sure, and the authors could recognize the level spacing, thus whatever the geometry is, the statement of the 1D limit is questionable. Usually, 1D means the level spacing due to the confinement along the wire could not be distinguished, i.e., effectively a continuous sub-band. The comparison to the superconducting gap is somehow loss of fundamental. In addition, the authors kept the word “In the case of TS...” (Line 35), which is misleading. Again, they did not expect TS in the system at all, and the system was not assumed to meet the criteria of TS. So I suggest a clear statement rather than mentioning something confusing.

We thank the reviewer for this comment. We understand better now what the reviewer is concerned about.

First of all, we thank the reviewer for pointing to the factor of 2 for the coherence length in our device which we indeed missed. It should be $\xi = \hbar v_F / 2\Delta^* \approx 5 \mu\text{m}$.

Regarding the 1D limit; as we already answered to the reviewer in the first round, there is clearly a longitudinal confinement. We fully agree with the reviewer that in the purely electronic sense, the 1D limit means that the level spacing cannot be distinguished. That happens for ballistic conductors if the level spacing is smaller than kT . For a CNT at $T \sim 30\text{mK}$, it corresponds to a length of the device of the order of the millimeter, completely out of reach and much larger than the superconducting correlation length and electron coherence length. For semiconducting nanowires (for example InAs NW), the Fermi velocity is about 10 to 100 times less than in CNT, so reaching the 1D limit in this sense would require devices between 10 μm to 100 μm which is also out of reach or at least has not been realized.

Instead we consider the 1D limit for the superconducting proximity effect which requires the length L of the NW to be larger than the proximity induced superconducting coherence length. It would be suitable for the emergence of TS and possibly topological modes if our NW had a suitable spin orbit coupling. Therefore our device offers a good benchmark of a system that cannot host TS but would show all the requirements except

one for the possible emergence of TS. We hope this last part also clears the confusion regarding our phrasing “In the case of TS...”.

We have rewritten this part of the main text to clarify better the 1D limit we consider as well as hopefully making clearer why we think it is a good benchmark for prospect of realizing TS in 1D systems.

3. From the explanation of the authors, I noticed that one of the four superconducting contacts worked properly, providing a superconducting proximity effect to the nanotube. One of them is poorer, and could be used as a normal probe. The other two could not be used for transport measurement. Please specify the information for the readers, since the yield and reliability of this method is important, for the so-called ultra-clean nanotube.

We have added a sentence in the main text to specify this information. Note that we realized that we previously did not specify the temperature (36 mK) at which the experiment was performed. We have added this important information now.

7. Data already published in ref. 20. I found it hard to accept the reason stated by the authors, since 4 out of the 6 authors in the current work were co-authored in ref. 20. It is now clear that the authors did not specify that part of the data in the current work has been published.

We respect the reviewer’s opinion to not accept our explanation but we maintain our statement that it was an honest oversight. We feel that what truly matters however is that we have explained clearly why we do not see it as a dual publication, which the reviewer does not dispute in his reply. We have added a sentence in the main text so that readers are made aware that part of the data map of Fig2a was published in Ref 20, as the reviewer suggests : “note that part of this data map was published in ref [20] to discuss the compatibility of the CNT nanoassembly technique with superconducting contacts”.

REVIEWERS' COMMENTS

Reviewer #1 (Remarks to the Author):

The authors have addressed the issues during the second round review. They have modified the manuscript and added additional information. I recommend its publication in Nature Communications.